# Do Neural Networks Trained with Topological Features Learn Different Internal Representations?

**Sarah McGuire**                      MCGUI176@MSU.EDU
*Department of Computational Mathematics, Science, and Engineering, Michigan State University*
*Pacific Northwest National Laboratory*

**Shane Jackson**                      SHANE.JACKSON@PNNL.GOV
*Pacific Northwest National Laboratory*

**Tegan Emerson**[*]                     TEGAN.EMERSON@PNNL.GOV
*Pacific Northwest National Laboratory*

**Henry Kvinge**[†]                      HENRY.KVINGE@PNNL.GOV
*Pacific Northwest National Laboratory*

**Editors:** Sophia Sanborn, Christian Shewmake, Simone Azeglio, Arianna Di Bernardo, Nina Miolane

## Abstract

There is a growing body of work that leverages features extracted via topological data analysis to train machine learning models. While this field, sometimes known as topological machine learning (TML), has seen some notable successes, an understanding of how the process of learning from topological features differs from the process of learning from raw data is still limited. In this work, we begin to address one component of this larger issue by asking whether a model trained with topological features learns internal representations of data that are fundamentally different than those learned by a model trained with the original raw data. To quantify "different", we exploit two popular metrics that can be used to measure the similarity of the hidden representations of data within neural networks, neural stitching and centered kernel alignment. From these we draw a range of conclusions about how training with topological features does and does not change the representations that a model learns. Perhaps unsurprisingly, we find that structurally, the hidden representations of models trained and evaluated on topological features differ substantially compared to those trained and evaluated on the corresponding raw data. On the other hand, our experiments show that in some cases, these representations can be reconciled (at least to the degree required to solve the corresponding task) using a simple affine transformation. We conjecture that this means that neural networks trained on raw data may extract some limited topological features in the process of making predictions.

**Keywords:** Topological data analysis, neural representation similarity, neural stitching, representation learning

## 1. Introduction

For some problems, deep learning has proven to be extremely efficient at extracting features from raw data which can then (at least to a moderate degree) generalize to unseen data.

---

[*] T.E. holds joint appointments in the Department of Mathematics at Colorado State University and the Department of Mathematical Sciences at the University of Texas, El Paso

[†] H.K. holds a joint appointment in the Department of Mathematics at the University of Washington

However, there remain classes of problems where domain expertise suggests that additional feature preprocessing should lead to heightened ML model robustness. One increasingly popular such type of data preprocessing involves using tools from topological data analysis (TDA) to extract topological features from data. These topological representations are then used to train an ML model. This growing field is often summarized as topological machine learning (TML). For example, in the analysis of microscopy images in materials science it is known that the size, abundance, and shape of precipitates in a material has a direct effect on the resulting physical properties of that material (Kassab et al., 2022). While an ML algorithm might learn to extract such features from raw data, the use of TML-based methods can ensure that these domain-informed features are available when a model makes predictions.

While there are an abundance of individual studies where topological features have been used to effectively train ML-models, there are still gaps in the community's understanding of how training with such features impacts the resulting model. In this work we take a first step in this direction by investigating the similarities and differences of models trained with raw data and models trained with topological features. We focus on the question of how such model's internal representations differ. While topological representations generally bear little resemblance to their corresponding raw data, it is conceivable that a network might learn to extract some topological features naturally. It is also possible that topological representations contain information that is "orthogonal" to the representations learned by a model trained with raw data. We aim to begin answering such questions in this paper.

To measure differences between hidden activations, we leverage two common tools used to quantify the similarity of neural representations. The first is model stitching (Lenc and Vedaldi, 2015; Bansal et al., 2021). This technique evaluates whether the features extracted by the first $k$ layers of one frozen $n$-layer model $f_1$ can be reconciled with the last $n - k$ layers of another frozen $n$-layer model $f_2$ after attaching them with a single learnable layer $L$. Informally, this method assumes such a composite network will be able to perform at least as well as either $f_1$ or $f_2$ if the representation of data learned by these models at layer $k$ is the same up to a transformation of type $L$. Note that $L$ is usually restricted to a specific family of relatively simple transformations (e.g., affine transformations, orthogonal transformations, or permutation matrices (Godfrey et al., 2022)). In this work we restrict ourselves to affine transformations. We also explore the differences in neural representations using centered kernel alignment (CKA), a metric that compares covariance matrices of hidden activations of networks (Kornblith et al., 2019). We argue that neural stitching measures task-focused similarity (can the information required for network $f_2$ to solve a task be extracted from the hidden activations of network $f_1$), while CKA measures more task-agnostic structural information.

We compare these representations for convolutional neural networks and multi-layer perceptrons (MLPs) trained on two different datasets (topologically distinguishable subsets of MNIST (Deng, 2012) and a subset of the describable textures dataset (Cimpoi et al., 2014)). In each case, one set of networks is trained on the raw images and one set is trained on persistence images (PIs) (Adams et al., 2017), a stable representation of the features in a persistence diagram. We draw several conclusions from our experiments. (1) We find that from a structural perspective, the representations learned by networks trained with persistence images are indeed distinct from those trained with raw images. On the other

hand, we find that to a limited degree (and depending on the network, dataset, and layer), the internal representations of models trained on persistence images can be used by the later layers of a model trained with raw data in order to solve a specific task. We argue that this implies that either a small amount of non-topological data is leaking into our persistence image representations or networks trained with raw data are actually preserving topological features.

In summary our work contains the following contributions:

- We use neural similarity metrics to quantify the relationship between the representations learned by neural networks trained with topological features and those trained on raw data.

- We show that the representations learned by networks trained with topological features differ considerably from those trained with raw data at a structural level.

- Despite these differences however, we show that from the perspective of solving a given classification task, these learned representations share some significant similarities.

## 2. Similarity metrics for neural representations

One approach to understanding how neural networks process data explores similarities and differences between the internal representations of distinct deep learning models. When such metrics are effective, they can reveal the answers to questions such as whether all models with strong performance tend to learn similar representations of a dataset regardless of random initialization or architecture (sometimes known as the 'Anna Karenina' scenario). Because of the value of such insight, considerable work has been put into identifying principled metrics of neural representations in both the deep learning and neuroscience literature (Kriegeskorte et al., 2008; Kornblith et al., 2019). These include: canonical correlation analysis (CCA) (Hardoon et al., 2004), along with its variants SVCCA (Raghu et al., 2017) and PWCCA (Morcos et al., 2018). All of these methods directly compare the data matrices obtained by extracting the hidden activations of a batch of input at some intermediate layers of two neural networks.

The two metrics that we chose to use in this paper are model stitching and centered kernel alignment (CKA). As described in the Introduction, model stitching (Lenc and Vedaldi, 2015) evaluates whether the hidden representation of one model can be transformed into a form that can be used by the later layers of another network to solve a task. Important conclusions about the way that deep learning models learn were later obtained using model stitching in (Bansal et al., 2021).

Let $f_1, f_2 : X \to Y$ be two models with input space $X$ and output space $Y$. Decomposing $f_1$ and $f_2$ into layers we can write: $f_1 = f_1^{n_1} \circ f_1^{n_1-1} \circ \cdots \circ f_1^1$ and $f_2 = f_2^{n_2} \circ f_2^{n_2-1} \circ \cdots \circ f_2^1$. We can stitch $f_1$ and $f_2$ together at layers $k_1 < n_1$ and $k_2 < n_2$ by creating

$$st(f_1, f_2, k_1, k_2) = f_2^{n_2} \circ f_2^{n_2-1} \circ \cdots \circ f_2^{k_2+1} \circ l \circ f_1^{k_1} \circ \cdots \circ f_1^1$$

where $f_2^{k_2+1}$ has domain $\mathbb{R}^{d_2^{k_2}}$, $f_1^{k_1}$ has range $\mathbb{R}^{d_1^{k_1}}$, and $l : \mathbb{R}^{d_1^{k_1}} \to \mathbb{R}^{d_2^{k_2}}$ is some layer belonging to a specified function class (e.g., affine transformations, linear transformations, or orthogonal transformations). We keep all weights in $st(f_1, f_2, k_1, k_2)$ except for those

in layer $l$. We then train $st(f_1, f_2, k_1, k_2)$ on the training set and evaluate its accuracy on the test set (we call this the *stitching accuracy*). If $st(f_1, f_2, k_1, k_2)$ is able to achieve performance comparable to either $f_1$ or $f_2$ then we draw the conclusion that a transformation of type $l$ is all that is need to reconcile the representations of $f_1$ at layer $k_1$ and $f_2$ at layer $k_2$ respectively.

Since our concern is for the effect of input type (topological vs non-topological), when we run stitching experiments, we record the accuracy when stitching together a model trained with topological input and a model trained with raw input. We then compare this with the result of stitching together two distinct models both trained with topological input or both trained with raw input. We take a greater difference in accuracy to mean a greater difference in internal representation between these two types of models (models trained with topological features and models trained with raw input). Note that this is a very task-centric approach to comparing representations, it measures similarity between representations based on how well one representation can be used by another network to complete a task.

On the other hand, centered kernel alignment (CKA) (Kornblith et al., 2019) take a more structure focused approach to measuring the similarity of representations in neural networks. CKA is defined as

$$\mathrm{CKA}(f_1^{<k_1}(D), f_2^{<k_2}(D)) = \frac{||\mathrm{Cov}(f_1^{<k_1}(D), f_2^{<k_2}(D))||_F^2}{||\mathrm{Cov}(f_1^{<k_1}(D), f_1^{<k_1}(D))||_F ||\mathrm{Cov}(f_2^{<k_2}(D), f_2^{<k_2}(D))||_F}$$

where $f_1^{<k_1}$ are the first $k_1$ layers of network $f_1$, $f_2^{<k_2}$ are the first $k_2$ layers of network $f_2$, $|| \cdot ||_F$ is the Frobenious norm, and $D$ is a set of samples from some data distribution. It is clear that CKA does not take into account task specific features of $f_1^{<k_1}(D)$ and $f_2^{<k_2}(D)$. That is, CKA can potentially take into account any structure in $f_1^{<k_1}(D)$ and $f_2^{<k_2}(D)$, whereas neural stitching implicitly focuses on structure that leads to good or poor performance in the second model in the stitching.

## 3. Topological features for machine learning

Topological data analysis (TDA) presents a collection of tools to characterize the shape of data. In particular, persistent homology quantifies shape by identifying holes in different dimensions: connected components, cycles, voids, and higher dimensional cycles. Topological information such as persistent homology representations have been incorporated into various aspects of ML problems: data pre-processing, feature extraction, model architecture, and training procedures.

There exists a collection of work which aim to extract topological features from data (for example, via persistent homology) and transform the representation for use as input to machine learning models. A primary subset of such work is different vectorization methods to transform topological features (which exist in spaces with properties such as non-unique means) into fixed-dimensional feature vectors, which are then suitable for machine learning. Vectorization methods include Betti curves, persistence images (Adams et al., 2017), persistence landscapes (Bubenik, 2015), silhouettes (Chazal et al., 2014), extracting signatures from point descriptors (Carrière et al., 2015), and other kernel-based vectorization methods (Reininghaus et al., 2015; Carrière et al., 2017; Kusano et al., 2018).

Additionally, there is a growing effort to incorporate topological information directly into machine learning models to influence or understand the architecture itself. One access point is regularization terms on the loss functions to encourage latent space representations to have certain topological features. Adjusting loss terms and regularization terms to account for topological information allows control over connectivity of an autoencoder's latent space such that topological features are preserved in latent space representations (Hofer et al., 2019; Moor et al., 2020). Additional applications of TDA have been used to understand and evaluate model behavior in different contexts: convolutional neural networks (Gabrielsson and Carlsson, 2019), generative adversarial networks (Zhou et al., 2021), classifier decision boundaries (Ramamurthy et al., 2019), and graph structure of neural networks (Rieck et al., 2019).

Recent surveys (Hensel et al., 2021; Barnes et al., 2021; Pun et al., 2022; Rabbani and Nugroho, 2020) provide more comprehensive discussion on topological information incorporated into machine learning and deep learning methods. These existing methods explore ways to leverage topological features (as primary representations, or to augment more standard representations) or use topology to understand topological information encoded in weights of the network. However, it is not well understood how training models with topological features actually affect the latent space representations of the model.

## 4. Experimental set-up

We chose to restrict ourselves to image datasets in this limited study. We strongly recommend examining these questions for other data modalities in future work. We were further constrained by the fact that we needed to choose datasets on which both a deep learning model trained with TDA and a deep learning model trained on raw features would perform reasonably well. For this reason, we used two simple datasets:

- **MNIST-0-1-8**: A subset of the standard MNIST dataset containing only the classes of topologically distinct digits '0', '1','8',

- **Describable textures BBV**: A subset of the *describable textures dataset* (Cimpoi et al., 2014) containing only the classes 'banded', 'bubbles', and 'veined', using the first split of data used for evaluation of the original dataset. We note that this is a challenging dataset for all models because of the few images available.

On these datasets we trained two types of models: those that take persistence images as input and those that take the raw images as input (see Figure 1). Where possible, we made changes to both raw images and PIs so that their size and number of channels were equal. We provide additional information on data preprocessing in A.

CKA and neural stitching are generally used to compare the representations of different models/layers which take as input the same data. In our experiments, the two models actually take as input different data representations (raw and topological). To reconcile this, we interpret the process of transforming a raw image into a persistence image as an initial layer in a model. Thus, when comparing representations, the underlying raw data is the same between both models.

We use three different model architectures: (i) an MLP with 4 blocks, batchnorm, and ReLU activations, (ii) a simple CNN with 6 blocks, and (iii) the standard AlexNet

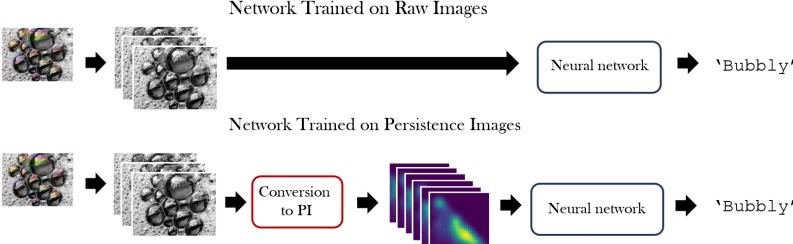

Figure 1: A layout of the two types of models we use in this paper (those taking as input raw images and those taking as input persistence images). We treat the process of calculating persistence images as a layer in the model to accord with standard practices when applying neural stitching and CKA.

architecture (exclusively for describable textures BBV). We trained our models on a 12GB NVIDIA Tesla P100 GPU with access to 16 cores and 64GB of memory. We provide training hyperparameters in Section B. All models were trained with standard cross entropy and the Adam optimizer (Kingma and Ba, 2014).

We choose to work with persistence images (PIs) (Adams et al., 2017) as our vehicle for extracting topological features from images. When computing PIs for MNIST-0-1-8, we use the raw single-channel images as input to the TDA pipeline. Raw images in Describable textures BBV, however, are RGB images and as such require separate PI computation for each of the three channels.

In our stitching experiments we connect linear layers with an additional linear layer (with appropriate dimension and a learned bias term). We stitch convolutional layers together with an 2$d$-convolutional layer with $1 \times 1$ kernel (and no bias term). Further exploration with different types of stitching transformations would be a valuable avenue of study. Stitching is often used to compare representations at different layers of a model (or representations at different layers of different models). Since we are already varying the input (persistence image vs. raw image) we always stitch together networks with identical architecture and always stitch at the same layer in both networks. That is, if both networks have $n$ layers, and we are using the first $k$ layers of the initial model $f_1$, then we always use the latter $n - k$ layers of model $f_2$. Analogous conditions hold for our CKA measurements.

We include 95% confidence intervals to indicate the amount of statistical variation among independently initialized and trained models. Our confidence intervals when stitching together a model trained with persistence images and a model trained with raw features are calculated over 4 distinct model pairs. Our confidence intervals when stitching together models trained with the same data type are calculated over 3 distinct pairs.

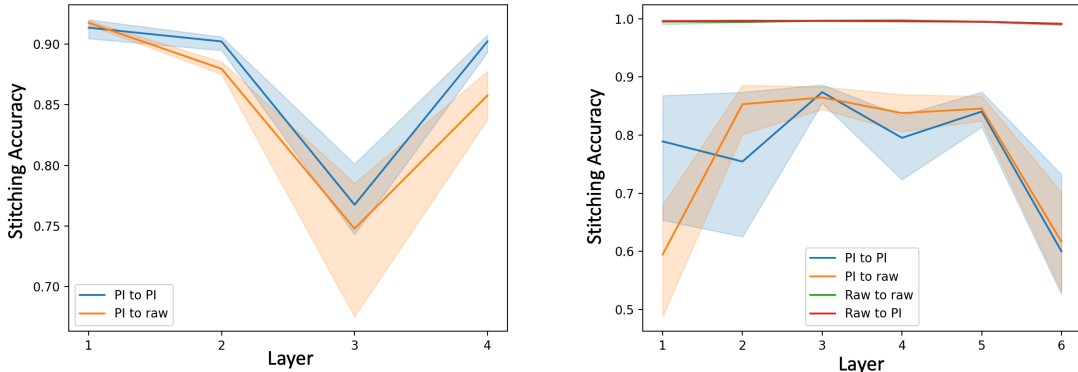

Figure 2: The stitching accuracy calculated when **(left)** MLPs (respectively CNNs **(right)**) trained on MNIST-0-1-8 are stitched together. The $x$-axis indicates the layer or block at which the networks were stitched. (Blue) corresponds to two networks trained on persistence images stitched together, (orange) corresponds to networks trained on persistence images (early) stitched to models trained on raw images (late), (green) corresponds to networks trained on raw images (early) stitched to models trained on persistence images (late), (red) corresponds a two networks trained on raw images. Shaded regions indicate 95% confidence intervals calculated over distinct pairs of randomly initialized and trained models (see Section 4 for details).

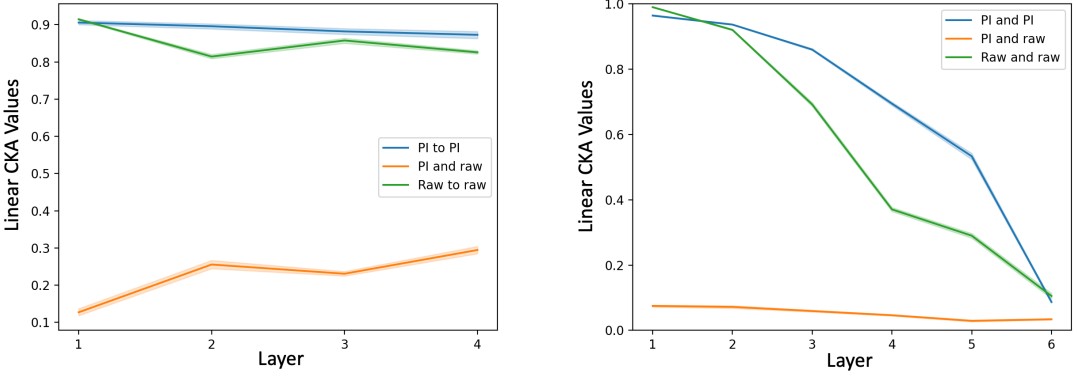

Figure 3: **(Left)** Linear CKA values computed for hidden activations of (blue) two MLPs trained on persistence images, (orange) an MLP trained on persistence images and an MLP trained on raw images, and (green) two MLPs trained on raw images. All networks were trained and evaluated on MNIST-0-1-8. **(Right)** The same experiment but with CNNs rather than MLPs. Shaded regions indicate 95% confidence intervals calculated over distinct pairs of randomly initialized and trained models (see Section 4 for details).

## 5. Results

### 5.1. Models trained with persistence images and models trained with raw images have structurally different representations

As noted in Section 2, CKA is one approach to measuring the structural similarity between the hidden activations of two different models. In Figures 3 and 4 (right) we show the result of applying CKA to representations from one model trained with persistence images and another trained with raw images (orange in both plots). For comparison, we also show the result of applying CKA to representations where both models were trained on the same type of input (persistence images in blue and raw images in green). Perhaps unsurprisingly, we see that the representations of models trained and evaluated on different datatypes (persistence images vs. raw images) are generally more dissimilar than the average representations of models trained on the same datatype.

Intriguingly, we note that in the case of MLP architectures the similarity between representations actually increases as data passes to deeper layers (though still remaining low overall). This suggests that these network's representations may actually converge to a certain extent at deeper layers of the model. On the other hand, this type of behavior is not seen in either the simple CNN or the AlexNet models where similarity begins to drop off in later layers (this is true for other types of CKA comparisons, not just comparisons between models using different data types). This points to a general phenomenon that we observe: the architecture used has a significant effect on the similarity trends observed from layer to layer.

### 5.2. Internal representations share more similarities when compared in the context of the task

Our neural stitching results tell a somewhat different story. We see that while for most models and most layers, stitching together models trained with different input types (topological vs raw) resulted in lower stitching accuracy than stitching together models trained on the same datatype, there were exceptions and in general the differences were much less significant than those captured by the task-agnostic CKA. For example, in Figure 2 we see that at intermediate layers of the CNNs trained on MNIST-0-1-8, stitching together a model trained on persistence images (earlier layers) and a model trained on raw images (later layers) actually led to statistically equivalent performance to stitching together two models trained on persistence images. It is unclear what is significant about the layers where this happens. Similarly, in Figure 4 we see that stitching together a model trained on persistence images (earlier layers) and raw images (later layers) can sometimes result in significantly higher accuracy than stitching together two different models trained on persistence images. This is the case for layer 4 in Figure 4.

It is important to note that our analysis is complicated by the fact that in general, the networks trained on raw images performed better than the networks trained on persistence images. This was especially true for the AlexNet models trained and tested on Describable Textures BBV, but also (to a lesser extent) for the MLP and CNN trained on MNIST-0-1-8. The difference in performance may be attributable to the fact that topological features tend to be significantly more coarse than standard raw input. It is also likely that AlexNet is

a suboptimal architecture to use for tasks involving persistence images since it has (approximate) translation invariance built into it (a feature which is beneficial to many vision tasks but not beneficial to making predictions on persistence images). However, it is still surprising that when stitching two networks together, those that were trained on different types of data would perform as well as those that were trained on the exact same data. It may be that despite a change in data type, the seemingly more robust feature extraction and processing learned by the networks trained with raw images provides a leg up when stitched to the early layers of a model trained on persistence images. It is also likely that the data itself plays a large role (this may explain the high stitching accuracy for red and green MNIST CNN models in Figure 2 that when stitched together take as input the raw MNIST images).

This phenomenon suggests a few possibilities. The first is that the way that models trained on raw images organize their internal representations can be closely approximated using topological information alone (at least enough to solve the task and achieve good stitching accuracy relative to the original models). The second is that in practice, topological representations leak a significant amount of non-topological features (this has already been shown to some extent in Turkeš et al. (2022)) which are used to reconcile the representations between models trained with persistence images and those trained on raw data. Otherwise, it is hard to conceive how stitched networks would be able to recover much of their performance via a simple learned stitching layer. Either way, there is clearly more to learn about how topological features interact with non-topological ones in deep learning models, both implicitly and when they are explicitly incorporated.

## 6. Limitations

There are a wide range of approaches to TML. While we investigated several axes representing the variation appearing between works in this field (e.g., model architecture and dataset) a more comprehensive set of experiments would be desirable. For example, one key area that we did not investigate was the use of different topological representations (we focused exclusively on persistence images). It would be useful to not only understand how neural representations differ between networks trained with and without topological features, but also between networks trained with different types of representations of features.

## 7. Conclusion

In this work we made a first step toward understanding how the neural representations of deep learning models trained with topological features differ from those trained with standard features. We find that while training with topological features does result in a network learning fundamentally different features, in certain cases (namely at the intermediate layers of a network) these representations contain enough common information that an affine transformation is enough to translate between one representation and another to the extent that a task can be solved. Through the lens of task-focused (neural stitching) and task-agnostic (CKA) neural similarity metrics, we are able to quantify the relationship between representations learned by neural networks trained with topological features, and those trained without.

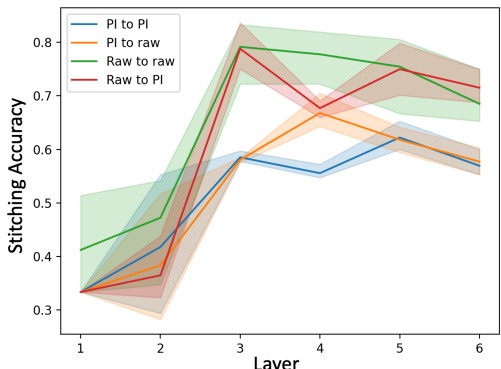 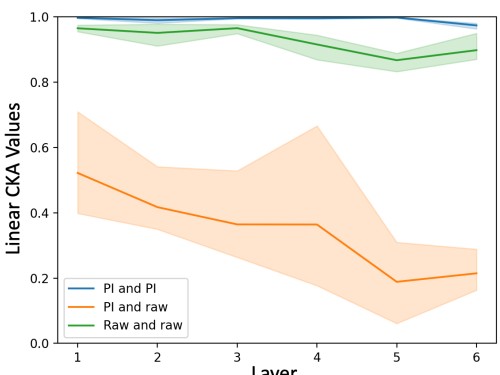

Figure 4: Results for AlexNet models trained the Describable Textures BBV dataset. **(Left)** The stitching penalties for (blue) two AlexNets trained on persistence images, (orange) an AlexNet trained on persistence images (early layers) and an AlexNet trained on raw images (later layers), (green) two AlexNets trained on raw images, and (red) an AlexNet trained on raw images (early layers) and an AlexNet trained on raw images (later layers). **(Right)** The corresponding measurement of CKA values. In both plots, the $x$-axis indicates where a network was either stitched or CKA values calculated. Shaded regions indicate 95% confidence intervals calculated over distinct pairs of randomly initialized and trained models (see Section 4 for details).

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

## Appendix A. Data preprocessing

In this section we give some further details on how we preprocessed images when training and evaluating our networks.

**Raw Images**: We transform grayscale MNIST images into RGB images to make them more comparable to persistence images which have at least 2 channels. We resized images to $32 \times 32$ before using them as input to our networks.

**Persistence Images**: We use the lower-star filtration of each grayscale image and compute 0-dimensional and 1-dimensional persistent homology. For color images (Describable textures BBV), the lower-star filtration is separately applied to each channel, resulting in separate persistent homology computations for each image channel. However, for grayscale images (MNIST-0-1-8), there is a single channel on which to apply the lower-star filtration. After persistent homology computations, each persistence diagram is vectorized into a persistence image with resolution $28 \times 28$, using weight parameter $n = 3.0$ and kernel parameter $\sigma = 0.003$. For MNIST-0-1-8, we append the 0-dimensional homology PI with the 1-dimensional homology PI, resulting in a 2-channel ($28 \times 28 \times 2$) image representation for each MNIST sample. For Describable textures BBV, we append the 0-dimensional homology PI with the 1-dimensional homology PI for each channel, resulting in a 6-channel ($28 \times 28 \times 6$) image representation for each texture sample. All persistence diagrams and subsequent persistence images are computed using `Ripser` (Bauer, 2021; Tralie et al., 2018) and `Persim` packages in `Scikit-TDA` (Saul and Tralie, 2019). We resized images to $32 \times 32$ before using them as input to our networks.

## Appendix B.  Hyperparameters

We provide the hyperameters used for training models in this work in Table 1. We provide the hyperparameters used when computing stitching accuracy in Table 2.

Table 1: Hyperparameter choices used when training models. MNIST denotes MNIST-0-1-8 and DTD denotes Describable Textures BBV.

|  | CNN/MNIST | MLP/MNIST | AlexNet/DTD-BBV |
|---|---|---|---|
| Initial learning rate | $5 \times 10^{-3}$ | $5 \times 10^{-3}$ | $5 \times 10^{-5}$ |
| Batch size | 64 | 64 | 12 |
| Weight decay | $10^{-5}$ | $10^{-5}$ | $10^{-5}$ |
| Training iterations | 2000 | 1000 | 2000 |

Table 2: Hyperparameters used when calculating stitching accuracy.

|  | CNN/MNIST | MLP/MNIST | AlexNet/DTD-BBV |
|---|---|---|---|
| Initial learning rate | $10^{-4}$ | $10^{-4}$ | $5 \times 10^{-5}$ |
| Batch size | 64 | 64 | 12 |
| Weight decay | $10^{-5}$ | $10^{-5}$ | $10^{-5}$ |
| Training iterations | 300 | 300 | 500 |

## Appendix C.  Model architectures

In this section we review the features of the two custom architectures used for the experiments in this paper. The AlexNet architecture that we used was drawn directly from Torchvision (Marcel and Rodriguez, 2010):

**MLP:** The MLP model used in this paper has 5 linear layers separated by ReLU nonlinearities.

**CNN:** The CNN model used in this paper has 6 convolutional blocks. Each block contains a $2d$-convolution, a batchnorm, and a ReLU nonlinearity. The model also includes 4 distinct pooling layers and a final linear layer for classification.

