# OpenReview forum: "Do Neural Networks Trained with Topological Features Learn Different Internal Representations?"
_NeurIPS.cc/2022/Workshop/NeurReps — NeurReps 2022 Oral_

### Official Review · Reviewer_9Xua · 2022-10-10
**Review of the paper "Do Neural Networks Trained with Topological Features Learn Different Internal Representations?"**

**Confidence:** 4
**Soundness:** 3
**Presentation:** 3
**Contribution:** 2
**Overall Rating:** 5

**Summary:**

In this paper, the authors investigated the difference between Deep Neural Networks (DNNs) trained with topological features and raw data. They exploited neural stitching and central kernel alignment (CKA) in two different datasets (MNIST 0-1-8 and describable textures BBV) with two different architectures (MLPs and CNNs). The paper represents an early-work investigating how models trained with topological features differ from those with raw data, which is highly relevant for the field of Topological Machine Learning.

**Questions:**

I have some doubts about the evaluation of the neural stitching. What do the plots refer to? The accuracy of stitching the two models or the penalty from stitching? How is it evaluated (a formula can be included in the appendixes)?

How are the stitching and CKA related? If I understood correctly, the CKA measures the similarity of the representations in the first $k_1$ and $k_2$ layers of the two DNNs respectively, while neural stitching measures the "compatibility" of the extracted features between the two models, modulo an affine transformation. Why stitching is high in certain layers and CKA always low?  Should high sticking imply that also the correlation is high?

**Limitations:**

Limitations are discussed throughout the paper and in a specific section. I found very clear the assessment of the authors in this regard.

**Recommended Decision:**

3: Accept

**Relevance:**

3: Solid fit

**Strengths And Weaknesses:**

$\large \textbf{Strenghts}$

The paper is well-written and easy to follow. Along the text, the authors stressed the importance of evaluating the difference between using topological input features (in their case, with persistent images - PI) and raw features and suggested a more in-depth evaluation should be carried out. The results on neural stitching for MNIST suggest that some correspondence between mid-level features of PI and raw features exist, which surprisingly leads to better performance with respect to stitching two equivalent PI-DNNs. This phenomenon holds also for two raw-DNNs in the BBV dataset. Conversely, the CKA highlights a substantial difference between the features extracted by the two models in all datasets.

$\large \textbf{Weaknesses}$

Despite these results already suggesting an interpretation of the difference between DNNs with raw features and PI features, the experimental investigation is somehow limited. This is an inherent limitation of the paper also acknowledged by the authors. Interestingly, the two metrics adopted show different behaviors: in my opinion, it would be a great improvement to explain further to what aspects these metrics relate to and compare them. See also the attached questions.

I spotted some typos: in the CKA formula, there are double parentheses. It should be $Cov(f_i^{<k_i}(D), f_j^{<k_j} (D))$, is that correct? Also, the citation "Kingma and Ba (2014)" in Chapter 4 seems in the wrong format.  As I understood $|| \bullet ||_F$ denotes the Frobenius norm of the matrix, it should be said somewhere in the text.


**Submission Track:**

Proceedings Paper (9 Page)

---

### Official Review · Reviewer_6S7T · 2022-10-15
**A solid paper with clear exposition about an Interesting research direction.**

**Confidence:** 4
**Soundness:** 3
**Presentation:** 4
**Contribution:** 3
**Overall Rating:** 7

**Summary:**

This work explores the changes training with topological features brings to the latent space of neural networks. The analysis is performed both extrinsically via model stitching and intrinsically via centered kernel alignment (CKA). The former measures if the two spaces considered are similar enough to be reconciled by means of a simple affine transformation (a trainable dense layer) to solve a downstream task, while the latter focuses on their geometry, measuring how well a correspondence can be established. The results show that even if the topological features determine a different shape of the latent space with respect to the one built from raw data (low CKA), a simple affine transformation is often enough to bridge the two (good task performance after stitching).

**Questions:**

- How many runs (with different random seeds or other sources of diversification) were done for each experiment? I was not able to find details about it in the text.
- What is the meaning of the area around the curves of grouped runs in figures 2, 3, and 4? Is it the standard deviation?
- Could the author please elaborate a little more on `This phenomenon suggests that either networks trained on raw images are already using topological features or that non-topological features leak into topological representations`? I find this interesting because, personally, I don't entirely agree with this statement. The reason being if an affine transform is enough to reconcile two latent spaces, it doesn't necessarily imply that they are built from (partially) shared features. Consider works that learn a mapping between latent spaces of different models (https://proceedings.neurips.cc/paper/2021/hash/01ded4259d101feb739b06c399e9cd9c-Abstract.html
 ) or completely different modalities (e.g., text captions and images), or even word embedding trained with different techniques or languages (https://arxiv.org/abs/1309.4168
 ). Features there are clearly different given their nature; it's the data semantics that stays the same and enables this stitching property, and I'm inclined to think it is the case in this work too.

**Limitations:**

- Have the authors tried tasks different from classification (e.g., reconstruction)? It could be interesting to verify if compatible results are obtained across different tasks. That could open for consideration about the data itself, reinforcing that it is not a phenomenon related to a specific task but a more general one.
- Have the authors considered other task-agnostic metrics or performed other studies of this kind? With CKA not giving good results in alignment, I wonder if it could just not be the best tool to measure the similarity at the geometry level in this setting.

**Recommended Decision:**

3: Accept

**Relevance:**

4: Highly relevant

**Strengths And Weaknesses:**

- A better understanding of the impact on the internal representations of NNs when using topological data analysis opens many possibilities (e.g., model ensembling, added interpretability, as well as forms of data augmentation). Therefore, I think the question posed by this work is stimulating for the community and worth digging deeper into in follow-ups.

- Personally, I wasn’t extremely familiar with TDA at the implementation level, but the work guides the reader effortlessly through the whole process. From high-level intuitions justifying the work and the experimental setup calling back to them, to low-level details about the implementations in the appendix. I particularly appreciated the effort in solving possible ambiguities and the stress about the duality of the experiments to help the reader understand the nature of the whole work.

**Submission Track:**

Proceedings Paper (9 Page)

---

### Official Review · Reviewer_rCyu · 2022-10-18
**An interesting, well-written exploratory study that encourages further investigation**

**Confidence:** 4
**Soundness:** 3
**Presentation:** 4
**Contribution:** 3
**Overall Rating:** 7

**Summary:**

The authors provide intriguing partial answers to the question of whether a neural network trained with topological features extracted from raw data learns representations of the data that are markedly different from those learned by a network trained on raw data, where the degree of difference is measured in terms of either neural stitching or centered kernel alignment (CKA).   They show in particular that while according to these measures the representations learned in the two contexts differ significantly, the representations can in some cases be reconciled by a simple affine transformation.

After a clear explanation of the similarity metrics that they apply, the authors briefly recall the foundations of TDA, in particular how one extracts topological features of data for machine learning.  They then describe their experimental set-up, involving classification tasks for two datasets (MNIST 0-1-8 and Describable textures BBV), for which neural networks (of three possible architectures) would be applied to either the raw data or associated persistence images, and clarify their stitching protocol.   They next carefully outline the results of their experiments, in which they exhibited significant structural differences (as measured by the CKA) between networks trained on persistence images and those trained on raw data.  Stitching tells a somewhat different, more subtle story: something odd seems to be happening in the middle layers, where a learned affine transformation seems to be able to correct for the difference between raw data and topological features.

**Questions:**

None

**Limitations:**

The authors express their awareness that sorts of topological representation other than persistence images should be tested as well and comparisons should be made between networks trained with different types of topological representations.

I think it would be interesting to consider as well raw data-types other than images and other architectures, such as attention networks.

**Recommended Decision:**

3: Accept

**Relevance:**

3: Solid fit

**Strengths And Weaknesses:**

Strengths: careful, well written, well justified conclusions

Weakness:  only an exploratory study, leaving much ground to cover.

**Submission Track:**

Proceedings Paper (9 Page)

---

### Decision · Program_Chairs · 2022-10-21

Accept (Oral)